# AMPLIFYING TRAINING DATA EXPOSURE THROUGH FINE-TUNING WITH PSEUDO-LABELED MEMBERSHIPS

## ABSTRACT

Neural language models (LMs) are vulnerable to training data extraction attacks due to data memorization. This paper introduces a novel attack scenario wherein an attacker adversarially fine-tunes pre-trained LMs to amplify the exposure of the original training data. This strategy differs from prior studies by aiming to intensify the LM's retention of its pre-training dataset. To achieve this, the attacker needs to collect generated texts that are closely aligned with the pre-training data. However, without knowledge of the actual dataset, quantifying the amount of pre-training data within generated texts is challenging. To address this, we propose the use of pseudo-labels for these generated texts, leveraging membership approximations indicated by machine-generated probabilities from the target LM. We subsequently fine-tune the LM to favor generations with higher likelihoods of originating from the pre-training data, based on their membership probabilities. Our empirical findings indicate a remarkable outcome: LMs with over 1B parameters exhibit a four to eight-fold increase in training data exposure. We discuss potential mitigations and suggest future research directions.

## 1 INTRODUCTION

Neural Language Models (LMs) have the ability to memorize extensive portions of their training data, which frequently encompasses sensitive information, thereby escalating significant privacy concerns. This is primarily due to *Training Data Extraction* (TDE) attacks (Carlini et al., 2021), enabling the disclosure of original training data during the model's inference phase. Numerous previous studies suggest that attackers, by generating extensive text and selecting outputs likely to contain training data, can access substantial amounts of sensitive information, even with restricted access to the model (Wallace et al., 2020; Carlini et al., 2021; 2023).

In this paper, we explore a novel attack strategy where a pre-trained LM is "adversarially" fine-tuned to increase the risk of exposing sensitive pre-training data. While many existing attack strategies emphasize post-hoc approaches (Lehman et al., 2021; Carlini et al., 2021; Balunovic et al., 2022; Carlini et al., 2023; Anil et al., 2023) to enhance the efficacy of TDE attacks against the fixed state of a target LM (*e.g.*, finding better prompts, modifying sampling methods, or developing ranking strategies), our approach probes the potential intensification of risk to the LM through the exploitation of self-generated text for fine-tuning. Our training objective also contrasts with existing defenses, such as differentially private training (Abadi et al., 2016; Anil et al., 2022) or self-distillation (Zhang et al., 2019; Tang et al., 2022), aiming to restrict the exposure of training data. The underlying assumption of our attack hinges on the availability of restricted white-box capabilities, which are becoming increasingly crucial and attainable, given the proliferation of public LMs (Scao et al., 2022; Touvron et al., 2023) and the advancement in black-box model extraction techniques (Carlini et al., 2020; Wu et al., 2023) (refer to §3.1 for more details).

Nonetheless, significant challenges remain. LMs may "forget" early training examples during the fine-tuning process (McCloskey & Cohen, 1989; Carlini et al., 2021; Jagielski et al., 2023), and ensuring the accuracy of labels on self-generated text is crucial for effective fine-tuning (He et al., 2019). To deal with these issues, attackers might employ self-generated texts that align with the original pre-training data, a process involving content quantification—akin to identifying texts with high membership. However, numerous TDE attack strategies necessitate empirical thresholds to differentiate between members and non-members of generated texts (Song & Mittal, 2021; Carlini

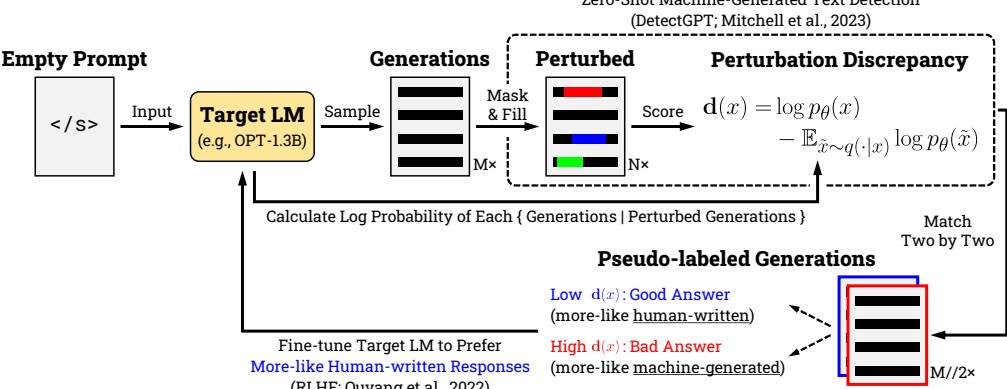

Figure 1: An overview. We feed an empty prompt into the target LM, consequently generating substantial text. For each piece of generated text, we calculate the perturbation discrepancy (Mitchell et al., 2023), where lower values signify a higher probability of the text being human-written and potentially containing sensitive training data. Subsequently, we match pairs of generations in twos and fine-tune (Ouyang et al., 2022) the target LM to favor the text with a lower perturbation discrepancy.

et al., 2022; Mireshghallah et al., 2022a;b). Without insights into the pre-training dataset, accurately determining membership becomes formidable, potentially leading to mislabeling texts that rarely contain training data as members.

To address these challenges, we propose the following two-fold strategy:

1. **Pseudo-Labeling based on Machine-Generated Probabilities** (§ 4.1): We generate extensive text from the target LM and pseudo-label (Lee et al., 2013) it, basing memberships on approximations. Utilizing the renowned zero-shot, machine-generated text detection method named DetectGPT (Mitchell et al., 2023), we infer machine-generated probabilities and inversely assign memberships. This method operates under the assumption that texts, even those machine-generated which incorporate training data, are likely to exhibit lower machine-generated probabilities.

2. **Reinforcement Learning with Self-Generations** (§ 4.2): We fine-tune the target LM utilizing its generated text. Employing Reinforcement Learning from Human Feedback (RLHF) (Christiano et al., 2017; Stiennon et al., 2020; Ouyang et al., 2022), which prioritizes relative sample preferences, we address the confirmation bias (Arazo et al., 2020) resulting from inaccurate labeling. This approach of pseudo-labeling prompts the target LM to favor responses reminiscent of training data.

We evaluate our approach using six distinct versions of the publicly available LM from the OPT (Zhang et al., 2022) family, namely 125M, 350M, 1.3B, 2.7B, 6.7B, and 13B. Consequently, our methodology enhances the efficacy of TDE attacks in fine-tuned LMs with over 1B parameters, exhibiting a four to eight-fold increase in effectiveness compared to reference LMs. Moreover, these fine-tuned LMs exhibit a heightened risk of exposing exceedingly lengthy sequences of training data, with instances including up to 1163 verbatim words.

In summary, our contributions are as follows: (1) We present a novel restricted white-box attack strategy (§ 3) amplifying the exposure of training data in LMs. This is achieved by pseudo-labeling of self-generated text (§ 4.1) and subsequent fine-tuning of the target LM (§ 4.2). (2) We provide empirical evidence supporting the feasibility of our approach, discerning pseudo-labeled membership discrepancies in generated texts (§ 5.2) and demonstrating its efficacy in amplifying training data exposure when targeting publicly available LMs (§ 5.3). (3) We delve into potential defensive strategies against our method, contributing valuable insights to the discourse on mitigating privacy risks posed to LMs (§ 6).

## 2 BACKGROUND

### 2.1 ZERO-SHOT MACHINE-GENERATED TEXT DETECTION

The proliferation of AI writing tools (Brown et al., 2020; OpenAI, 2023; Anil et al., 2023) has driven the need for detectors capable to determine whether a text is machine-generated. Among the extensive prior research addressing this issue (Zellers et al., 2019; Ippolito et al., 2020; Fagni et al., 2021), we focus on a popular zero-shot machine-generated text detection strategy called DetectGPT (Mitchell et al., 2023).

DetectGPT determines the machine-generated nature of a given text based on the expectation of its log probability. Specifically, the expectation of the difference in log probabilities between a given text and its perturbed texts—*e.g.*, replacing or deleting words; termed *perturbation discrepancy*—converges to 0 for machine-generated text. Otherwise, there is a higher likelihood that it is human-written. Formally, perturbation discrepancy is defined as (Mitchell et al., 2023):

$$\mathbf{d}(x, p_\theta, q) \overset{\Delta}{=} \log p_\theta(x) - \mathbb{E}_{\tilde{x} \sim q(\cdot|x)} \log p_\theta(\tilde{x}) \qquad (1)$$

where $x$ is the text we want to classify as machine-generated or not, $p_\theta$ represents the source model from which we want to discern if the text $x$ was derived, and $q(\cdot \mid x)$ is a function that generates a perturbed version using $\tilde{x}$ as the base, like T5 (Raffel et al., 2020).

### 2.2 REINFORCEMENT LEARNING FROM HUMAN FEEDBACK

RLHF (Christiano et al., 2017; Stiennon et al., 2020; Ouyang et al., 2022; OpenAI, 2023) is a fine-tuning strategy for LMs using human preferences as a reward signal. This strategy is divided into three sequential steps: First, fine-tune the target LM to produce the desired output for any given prompt as labeled by a human, termed Supervised Fine-Tuning (SFT). Secondly, rank the target LM's responses for the same prompt by labelers. After which, train a Reward Model (RM) to approximate these human preferences. Lastly, reinforcement learning, via Proximal Policy Optimization (Schulman et al., 2017) (PPO), is employed to search for the optimal policy—specifically, the parameters of the target LM—that maximizes the expected reward for the generated text. Through these three steps, the behavior of the target LM is aligned to reflect the intricate human preferences.

## 3 THREAT MODEL

In this section, we establish the adversary's capabilities (§ 3.1) and define the objective to achieve from the target LM through these capabilities (§ 3.2). An accurate threat model demonstrates the operating principle and limitations of our attack, which will also help design future defense strategies.

### 3.1 ADVERSARY'S CAPABILITIES

We consider a restricted white-box adversary who has access to the input-output of the target LM and can even fine-tune it as desired. This scenario permits access to the entire set of parameters of the target LM, but the distribution and attributes of the private training data are not allowed.

The assumptions of this restricted white-box setting are becoming increasingly essential and realistic for two reasons: (1) Pre-trained LMs, trained on private datasets or those of sizes inaccessible to an adversary, are gradually being open-sourced (Kim et al., 2021; Touvron et al., 2023; Rozière et al., 2023) due to the efforts of various organizations promoting open science, and (2) Adversaries can extract parameters of neural networks, including LMs, even in a black-box access environment (Carlini et al., 2020; He et al., 2021; Wu et al., 2023).

### 3.2 ADVERSARY'S OBJECTIVE

The adversary's objective is to maximize the amount of private training data exposed from the generated texts of the target LM, *i.e.*, true positive. Specifically, given a pre-trained LM $f_\theta$ with parameters $\theta$, the adversary wants to maximize the effectiveness of the TDE attack:

$$\text{Maximize} \quad \mathbb{E}_{\hat{y} \sim \mathcal{D}_{\text{infer}}} \Big[ \mathbf{1}_{\text{condition}} [\exists \, i : \hat{y}_{i..i+k} \in \mathcal{D}_{\text{train}}] \Big] \qquad (2)$$

where $\mathcal{D}_{\mathrm{infer}} = \left\{ \hat{y}^{(i)} \right\}_{i=1}^{l}$ is a set of $l$ sequences sampled from the LM $f_\theta$ by the adversary with the prompt $x$, $\hat{y} = [\hat{y}_1, \hat{y}_2, \cdots, \hat{y}_m]$ is a generated text consisting of $m$ tokens excluding the prompt $x$, and $\hat{y}_{i..i+k} = [\hat{y}_i, \hat{y}_{i+1}, \cdots, \hat{y}_{i+k}]$ is partial generations consisting of $k$ consecutive tokens starting from index $i \in [1, m-k]$. $k$ and $l$ are predefined hyperparameters.

This paper does not aim for a targeted attack intending to extract data with specific attributes. Rather, we consider an untargeted attack where input prompt $x$ is empty (*i.e.*, ``). The targeted attack that injects prompts[1] that are easily accessible to adversaries, such as personal identification information, medical information, or code snippets, can increase the effectiveness of the attack and be helpful in quantifying memorization (Carlini et al., 2023). However, it is not realistic as it mandates prior knowledge of the training data.

## 4    STUDY DESIGN

In this section, we describe our TDE attack strategy. First, we pseudo-label the generated texts based on the machine-generated probability (§4.1). We then utilize these pseudo-labeled texts to perform reinforcement learning (§4.2).

### 4.1    PSEUDO-LABELING BASED ON MACHINE-GENERATED PROBABILITIES

**Generating Texts.**    We first input an empty prompt, specifically "``," into the target LM to produce 100,000 texts (Carlini et al., 2021). By feeding the unique token that indicates the beginning of a sentence, we can extract the most confident samples from the LM. This method corresponds with the adversary's objective as texts generated with higher confidence are more likely to be memorized training data (Shokri et al., 2017). While there is no length constraint for the generated texts, we deliberately fixed the number of tokens of each generated text to 256 to enhance the reliability of the TDE attack performance. This length is consistent with that of texts after our TDE attack. Despite the possibility of duplicated data among the generated texts (Carlini et al., 2021), we opted not to carry out any deduplication for actual attack performance computation.

**Perturbing Generated Texts.**    Subsequently, we produce ten perturbed texts for each generated text using the mask-and-fill approach. We repetitively mask two consecutive spans until 15% of the words delineated by spaces are corrupted (Mitchell et al., 2023). To prevent the influence of each masked span within a single sentence from becoming excessively dominant, we dropped texts comprising fewer than 20 words. We used a T5-Large (Raffel et al., 2020) model, pre-trained with a span-corruption objective to predict masked spans.

Given the vast number of texts to produce perturbation from, we simplify the machine-generated text detection method proposed in prior work (Mitchell et al., 2023) for efficiency. For instance, while previously 100 perturbed texts were produced for each generation, we only produced ten and replaced the perturbation function from T5-3B (3B) to T5-Large (770M). While such a reduced perturbation may induce inaccurate labels, we minimized the confirmation bias by a trick in the subsequent pseudo-labeling step. For concrete examples of mask-and-fill on the generated text, please refer to Appendix C.1.

**Calculating Perturbation Discrepancy for Each Generated Text.**    Next, we compute the log-likelihood of the generations and perturbed texts respective to the target LM. As previously mentioned in §2.1, the perturbation discrepancy of a random generation from the target LM is calculated by the difference between that generation's log probability and the perturbed texts' expected log probability. Unlike previous studies that classified texts with perturbation discrepancy exceeding a threshold as machine-generated (Mitchell et al., 2023), we only compare the perturbation discrepancy between two generated texts; specifically, a lower discrepancy is assumed to more likely contain human-written text.

**Pseudo-Labeling Texts through Perturbation Discrepancy.**    Lastly, we pair two generated texts with their perturbation discrepancy. Note that the text preferred by the target LM is likely human-

---

[1]For instance, consider a scenario where the adversary injects the following prompt to extract specific email information: "If you have other issues, please contact us as"

written, meaning it would have a relatively lower perturbation discrepancy. Consequently, we can naturally determine the pseudo-label of membership (*i.e.*, chosen and rejected) by categorizing paired texts based on lower and higher perturbation discrepancies. For detailed examples of pseudo-labels determined by perturbation discrepancy within a pair, consult Appendix C.2.

The most trivial method to select such pairs is randomly mapping two texts. However, in this case, due to our simplified implementation (*i.e.*, reduced number of perturbed texts per generation and use of a perturbation model with low capacity), the reliability of the pseudo-label may be somewhat compromised. On the contrary, to find the globally optimal pair with a maximized difference in perturbation discrepancy, one must examine all possible combinations, which is computationally prohibitive due to its quadratic nature. As a compromise between the two solutions, we sort texts by perturbation discrepancy and sequentially select one from the top-scoring half and one from the remainder to match. A method that ensures the maximum discrepancy difference between pairs, like the heuristic algorithm of simulated annealing (Bertsimas & Tsitsiklis, 1993), could further enhance this, and remains a topic for future research.

## 4.2 REINFORCEMENT LEARNING WITH SELF-GENERATIONS

To fine-tune the target LM using the pseudo-labeled self-generation dataset, we apply the popular fine-tuning strategy for large LMs, called RLHF (§2.2). We modify the reward from human feedback to perturbation discrepancy to promote the target LM to favor responses expected to contain more training data.

We have omitted the SFT process for the target LM. The primary aim of SFT is to modify the response format of an LM trained with a causal language modeling objective to act like a chatbot by adding simple directives at the beginning and end of a prompt[2]. We assessed that this process is inconsistent with our attack strategy of exposing training data by inputting an empty prompt. Therefore, we used 40% and 60% of the pseudo-labeled dataset to fine-tune RM and the PPO algorithm, respectively. All other unspecified methods follow the approach of Ouyang et al. (2022). Please refer to Appendix B.3 for the detailed specifications of the fine-tuning dataset.

## 5 EXPERIMENTS

In this section, we experimentally validate the feasibility and effectiveness of our two-step approach in amplifying the exposure of training data by addressing the following two research questions:

- **RQ1 (Feasibility)**: Can the RM discern generated texts containing more training data by fine-tuning with pseudo-labeled texts based on the perturbation discrepancy difference? (§5.2)
- **RQ2 (Effectiveness)**: If discernible, can we amplify training data exposure by fine-tuning target LM using the trained RM? (§5.3)

RQ1 confirms the validity of our approach. Drawing on earlier studies that identified a perturbation discrepancy gap related to the presence of training data in text (Mitchell et al., 2023), we experimentally show that RM can differentiate between texts based on this discrepancy, achieving significant binary accuracy.

In RQ2, we quantitatively analyze whether our approach enhances the performance of TDE attacks based on the results of RQ1. We perform the same TDE attacks on the reference LM and fine-tuned LM, and observe the true positives of these attacks.

## 5.1 EXPERIMENTAL SETUP

**Settings.** We demonstrate our attack on the famous LM, OPT (Zhang et al., 2022), which includes model parameters and a public-available training dataset. Given that the OPT family contains nine different architectures ranging from 125M to 175B, it facilitates observing performance trends based

---

[2]For instance, for a prompt like "How are you?", one can add tokens indicating human and assistant directives to transform it into "Human: How are you? Assistant:".

Table 1: The test accuracy over epochs when fine-tuning the RM using datasets created from each OPT version. All experiments present the average and 95% confidence interval from five repeated trainings on different dataset splits. Epoch 0 denotes before RM's training starts.

| OPT | Epoch 0 | Epoch 1 | Epoch 2 | Epoch 3 |
|---|---|---|---|---|
| 125M | $49.7 \pm 1.1$ | $\mathbf{65.5 \pm 1.1}$ | $65.5 \pm 1.6$ | $63.9 \pm 1.8$ |
| 350M | $52.2 \pm 1.4$ | $69.4 \pm 2.3$ | $\mathbf{70.1 \pm 1.9}$ | $68.7 \pm 0.9$ |
| 1.3B | $51.2 \pm 2.2$ | $69.2 \pm 1.3$ | $\mathbf{69.2 \pm 1.3}$ | $67.9 \pm 0.9$ |
| 2.7B | $50.9 \pm 0.7$ | $66.1 \pm 2.3$ | $\mathbf{66.4 \pm 0.6}$ | $65.5 \pm 1.2$ |
| 6.7B | $50.5 \pm 1.8$ | $\mathbf{64.8 \pm 1.5}$ | $64.3 \pm 0.3$ | $62.8 \pm 1.0$ |
| 13B | $51.4 \pm 2.2$ | $\mathbf{62.9 \pm 0.7}$ | $62.6 \pm 0.9$ | $61.6 \pm 0.5$ |

on the LM scale. Due to limited experimental resources, we restrict our experiments to the following six versions of OPT: 125M, 350M, 1.3B, 2.7B, 6.7B, and 13B. All RMs are pre-trained OPT-350M. Please refer to Appendix B.4 and Appendix B.5 for fine-tuning RMs and the target LMs, respectively.

We fix all attack hyperparameters for text generation to observe the change in training data exposure through our fine-tuning strategy. We simultaneously use top-$k$ sampling—restricting sampling to the $k$ vocabulary tokens with the highest probability—with $k = 40$ and top-$p$ (Holtzman et al., 2019) sampling—restricting sampling to the smallest set of most probable tokens whose probabilities sum up to at least $p$, also referred to as nucleus sampling—with $p = 0.95$. To avoid repetitive phrasing, we ensure the same trigram appears no more than once within a generated text. All generated texts contain 256 tokens each, excluding the prompt. We do not employ temperature to flatten each token's probability.

**Verification.** To verify whether the generated texts from the target LM indeed contain training data, we consider twelve original training datasets of OPT: BookCorpus (Zhu et al., 2015), CC-Stories (Trinh & Le, 2018), Pile (Gao et al., 2020) (containing Pile-CC, OpenWebText2, USPTO, Project Gutenberg, OpenSubtitles, Wikipedia, DM Mathematics, and HackerNews), Pushshift.io Reddit dataset (Baumgartner et al., 2020), and CCNewsV2 (Liu et al., 2019). We used only ten datasets for verification, excluding CC-Stories, which is no longer available in its original form, and CCNewsV2 due to its massive size. We did not deduplicate each dataset, unlike their original pre-processing phase. Please refer to Appendix B.2 for the specific specifications of OPT's ten reconstructed pre-training datasets.

**Evaluation Metrics.** We evaluate the performance of our attack strategy as true positives per 100,000 generated texts. Following the convention for TDE attacks (Lee et al., 2022), we consider sentences as *extracted* when they have over 50 duplicated tokens from the original training data.

Computing the true positives of generated sentences by individually identifying overlaps within the ten datasets above demands high computational costs. Instead, we employ the suffix array (Manber & Myers, 1993)-based exact substring duplication (Lee et al., 2022) strategy—i.e., EXACTSUBTR—to search for duplicate text between generations and the ten original training datasets. This method can identify duplicated examples in linear time (Lee et al., 2022), and as the implementation is done in Rust (Matsakis & Klock, 2014) rather than Python, it guarantees very high computational speeds. Using this strategy, we report the number of entirely non-duplicated unique generated sentences.

## 5.2 RQ1: DISCRIMINABILITY OF PERTURBATION DISCREPANCY IN GENERATIONS

Note that RM is fine-tuned to award more rewards to the text with higher membership based on perturbation discrepancy among two different paired texts, *i.e.*, chosen and rejected. Thus, if the binary classification accuracy on a test dataset significantly exceeds 50%, the expected accuracy of random guessing, after sufficient training, we can argue that RM can capture the difference in perturbation discrepancy.

Table 1 displays binary classification accuracy based on RM's fine-tuning epoch for the same training and test datasets. To meticulously observe the trend of performance changes through fine-tuning, we deliberately induce overfitting using multiple epochs. This contrasts with the actual learning phase

Table 2: True positives of the TDE attack on 100,000 generated texts from the reference LM (●) and our fine-tuned LM (○). We did not conduct repeated experiments since we believe that generating 100,000 massive texts can reduce bias for true positives.

| OPT | [50, 64) ● | [50, 64) ○ | [64, 128) ● | [64, 128) ○ | [128, 192) ● | [128, 192) ○ | [192, 256) ● | [192, 256) ○ | 256 ● | 256 ○ | Total ● | Total ○ | Inc. |
|---|---|---|---|---|---|---|---|---|---|---|---|---|---|
| 125M | 64 | 54 | 24 | 80 | 5 | 20 | 8 | 15 | 0 | 0 | 101 | **169** | ×1.7 ↑ |
| 350M | 103 | 91 | 64 | 128 | 11 | 35 | 29 | 71 | 0 | 0 | 207 | **325** | ×1.6 ↑ |
| 1.3B | 58 | 241 | 38 | 337 | 0 | 52 | 1 | 139 | 0 | 6 | 97 | **775** | ×8.0 ↑ |
| 2.7B | 53 | 216 | 72 | 253 | 2 | 21 | 0 | 27 | 0 | 0 | 127 | **517** | ×4.1 ↑ |
| 6.7B | 87 | 174 | 57 | 220 | 1 | 53 | 0 | 98 | 0 | 0 | 145 | **545** | ×3.8 ↑ |
| 13B | 87 | 347 | 101 | 394 | 5 | 27 | 0 | 18 | 0 | 0 | 193 | **786** | ×4.1 ↑ |

by only running one epoch. We have also observed some flawed learning outcomes for multiple times, such as RM's test accuracy converging to 0 or neither training loss nor accuracy monotonically increasing. To minimize bias in experimental results, we present outcomes of repeated RM fine-tuning on different seeds until a valid result emerges five times. Consider that an adversary can also control RM's fine-tuning, hence repeating training until success is deemed reasonable. Through Table 1, we show the followings:

**Pre-trained RM cannot distinguish perturbation discrepancy differences between generations.** We have confirmed that the test accuracy for RM before training—*i.e.*, at epoch 0—is roughly around 50%. This randomness is essentially evident since we have not yet fine-tuned the RM based on the perturbation discrepancy differences.

**RM can learn about perturbation discrepancy differences between generated texts through fine-tuning.** A trained RM is more accurate than an untrained RM, indicating it can learn the perturbation discrepancy difference between generated texts as intended. RM also shows a slight decrease in test accuracy as the epoch increases when learning the pseudo-labeled dataset for some versions, which can be attributed to overfitting. For some results that showed the highest accuracy at epoch 1, it is possible to optimize the RM's performance by reducing the training dataset size.

**RMs that learned pseudo-labeled generations derived from larger LM show lower classification performance.** It is evident that an LM with more parameters is more likely to generate more realistic text; hence, the difference in perturbation discrepancies diminishes (Mitchell et al., 2023). This narrowed gap can be linked to a decline in the quality of the fine-tuning dataset endowed with pseudo-labeled membership. As an empirical evidence, we observed a gradually decreasing trend in the test accuracy of RMs trained on pseudo-labeled datasets derived from larger models. Considering high-performance perturbation functions—*e.g.*, T5-3B Raffel et al. (2020)—or generating more perturbed texts can be an approach to enhance RM's performance.

## 5.3 RQ2: Possibility of Amplifying Training Data Exposure via Fine-tuning

Subsequently, we examine the changes in the exposure of training data by applying RLHF to the target LM with RM which can distinguish differences in perturbation discrepancy. We observe the results by dividing the sufficiently large number of duplicate tokens of the generated text into five intervals: [50, 64), [64, 128), [128, 192), [192, 256), and {256}.

Table 2 shows the true positives of reference LM and fine-tuned LM, categorized by OPT versions and duplication intervals[3]. Through the experiment, we confirmed that our fine-tuning approach consistently boosts the training data exposure of the reference LM. The amplification of training data exposure is more pronounced in larger models, remarkably increasing up to 8 times in the OPT-1.3B.

---

[3]Due to various constraints, our results may not precisely match the true positives. We could not prepare all the training data for OPT; multiple duplicates can still exist in generated texts.

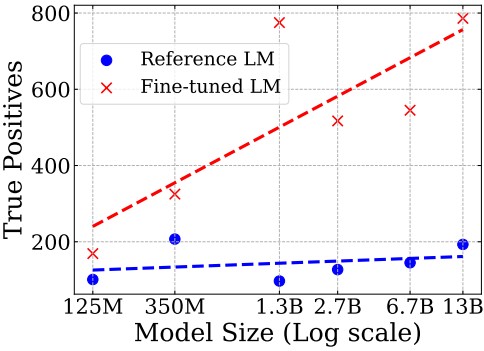

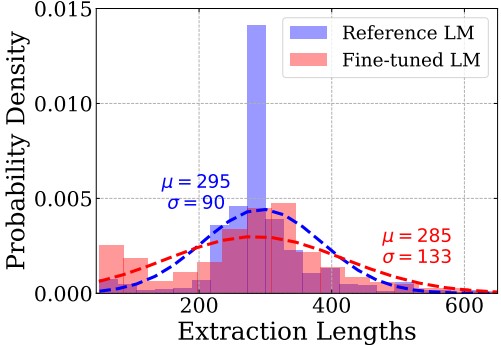

Figure 2: True positives by model scale for reference LM (blue ○) and fine-tuned LM (red ×). We also performed linear approximation (dotted line), and the confidence determination coefficient $R^2$ for the reference LM and fine-tuned LM are $0.65$ and $0.09$, respectively.

Figure 3: Distribution of verbatim text lengths extracted from the reference LM (blue) and the fine-tuned LM (red). The maximum lengths of training data extracted from the reference LM and the fine-tuned LM are $885$ and $1163$, respectively.

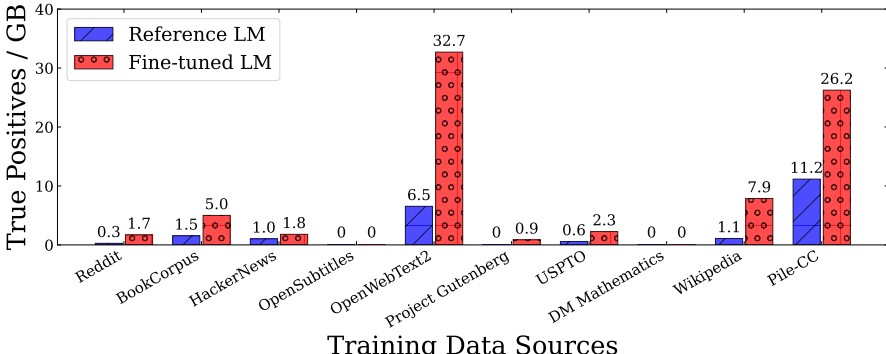

Figure 4: True positives per GB for OPT training datasets for reference LM (blue) and fine-tuned LM (red). A value of $0$ indicates that no training data was extracted from that dataset.

Furthermore, we present true positives by model size in Figure 2. The known fact from previous studies is that as the model size increases, the exposure of training data increases log-linearly (Carlini et al., 2023). We further demonstrate that our fine-tuning can rapidly accelerate this exposure.

## 5.4 QUALITATIVE ANALYSIS OF EXTRACTED SAMPLES

We perform a qualitative analysis of several attributes of samples extracted from either the reference LM or the fine-tuned LM. While OPT was trained with public-available datasets like the Pile (Gao et al., 2020), the types of sub-training datasets are very diverse, making the qualitative analysis of memorized content an intriguing subject.

**Training Data Sources.**    We investigated which training data the extracted training data belongs to. Since each reconstructed dataset has different sizes (see details in Appendix B.2), we calculated the true positives per GB for each dataset for a fairer comparison. Figure 4 shows the results. We could not extract any data from the OpenSubtitles and DM Mathematics datasets. We speculate that these two datasets might have resisted TDE attacks because of their relatively high complexity. Wikipedia is the most vulnerable dataset in TDE attack, which was increased about $7.2$ times after fine-tuning. Quantifying leakage levels according to each dataset type and measuring their risks are interesting directions for future work.

**Extraction Length.** Figure 3 displays the distribution of the verbatim length of the texts extracted from both the reference LM and the fine-tuned LM. Note that we count the length in words, not tokens. Training data leaked from the reference LM tends to have a similar average length, whereas the fine-tuned LM emits a relatively more comprehensive range of training data. The maximum lengths of training data extracted from the reference LM and the fine-tuned LM are $885$ and $1163$, respectively. Considering the true positives for each interval in Table 2, fine-tuning the target LM enables more extraction of longer texts.

## 6    Possible Mitigations and Countermeasures

So far, we have shown that if an attacker knows the entire parameters of the target LM, they can amplify training data exposure by fine-tuning the model with pseudo-labeled membership. A natural question arises on the possible mitigations and countermeasures against our amplifying exposure strategy. Since our approach introduces a novel type of TDE attack not previously reported, we discuss two potential defense strategies:

**Reducing Reliability of Machine-generated Probabilities.** Our TDE attack assumes that we can accurately compute the machine-generated probability of generated texts. Thus, reducing the reliability of this probability can lead to a decrease in the quality of the fine-tuning dataset, which can consequently reduce the effectiveness of the TDE attack. To reduce this reliability, we can consider two methods: (1) Reducing the statistical difference between machine-generated and human-written text distributions by making the LM more sophisticated and powerful (Sadasivan et al., 2023). However, merely increasing the size of the LM might inadvertently enhance memorization and thus boost the default performance of the TDE attack (Carlini et al., 2023). Therefore, enhancing the LM's performance while maintaining its scale would be effective. (2) Limiting the LM's training dataset to a domain that makes mask-filling of perturbation functions ineffective. For instance, DetectGPT showed notably lower detection capabilities on the PubMedQA (Jin et al., 2019), a biomedical research dataset crafted by experts, than on other typical datasets (Mitchell et al., 2023). An LM trained in non-English might also hinder the functioning of perturbation functions. Without knowledge of the training dataset, an adversary might be restricted from deploying adaptive attacks using multilingual models like mT5 (Xue et al., 2021).

**Fine-tuning to Reduce Training Data Exposure.** On the other hand, we can consider a strategy that involves flipping the pseudo-labels for membership, directing the target LM's fine-tuning towards reducing the training data exposure. Since defenders already have access to high-quality training data, they can easily adopt this approach. However, the impact of fine-tuning with pseudo-labeled self-generations on the generalized performance of the LM is a subsequent concern. The pre-trained model's generalizable representations could be degraded during the fine-tuning process, known as a *representation collapse* phenomenon (Aghajanyan et al., 2020). While attackers do not need to consider the target LM's performance—*e.g.*, validation perplexity—, defenders must balance privacy and utility. To efficiently counter privacy attacks without compromising the usefulness of the target model, we can consider a strategy like RelaxLoss (Chen et al., 2022) during the fine-tuning process that intentionally relaxes the target loss.

## 7    Conclusion

This paper presents a novel form of TDE attacks wherein a pre-trained LM is adversarially fine-tuned, enhancing the risk of exposing sensitive pre-training data. Given the recent exponential growth in LM parameters, our attack strategy raises serious concerns, as it tends to be more effective in larger LMs. We leave several open questions for promising future research: (1) How does fine-tuning with self-generated samples specifically affect the retention of memorization? (2) Can fine-tuning the target LM to favor responses with less training data genuinely contribute to mitigating TDE attacks? (3) Can our approach be extended beyond neural LMs to increase training data exposure in general generative models? By further exploring these open questions, we hope our work will contribute to enhancing the robustness of LMs and other generative models against TDE attacks.

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

The appendix begins on the following page.

# A  RELATED WORK

## A.1  DATA AUGMENTATION-BASED MEMBERSHIP INFERENCE

Membership Inference (Shokri et al., 2017) (MI) attacks try to determine whether a data point was part of the training dataset of a Machine Learning (ML) model. Overfitting is known to be a primary reason why training data becomes vulnerable to MI attacks (Yeom et al., 2018; Ye et al., 2022). Some previous studies infer the fitting status from the difference between the loss for an arbitrary data point and the expected loss for its augmentations; specifically, if the sample is a non-member, the expectation of the difference converges to 0, but if it is from the training dataset, it takes a distinguishably small negative value.

From this perspective, the directly related work to ours is by Mattern et al. (2023). They calculate the augmentation of the target sample using a masked LM like BERT (Devlin et al., 2018) and compare the expected loss difference, a process similar to our method of calculating perturbation discrepancy (as described in §4.1). However, their study differs from our approach: (1) They require a pre-calculated threshold to decide membership based on the difference, while we do not need a specific threshold. This difference makes our method more straightforward as we only compare relative membership. (2) Their attack, like many others, solely audits the vulnerability of the target model to MI, whereas we focus on enhancing this vulnerability to amplify the attack's impact.

## A.2  AMPLIFYING OTHER SECURITY CONCERNS IN LANGUAGE MODELS VIA FINE-TUNING

We examine various research on amplifying security concerns by fine-tuning LMs, including the training data exposure. Mireshghallah et al. (2022b) demonstrate that fine-tuning the LM can induce overfitting to memorization, making it susceptible to MI attacks. However, they focus only on the fine-tuning dataset, not the pre-training dataset, to claim the data becomes vulnerable to MI attacks through fine-tuning. This is significantly different from our fine-tuning strategy aimed at exposing the pre-training dataset. As another example, Shi et al. (2023) tried to inject a backdoor into an LM fine-tuned using PPO by poisoning (Biggio et al., 2012) the dataset during the RLHF process.

# B  IMPLEMENTATION DETAILS

## B.1  ENVIRONMENT SETTINGS

Due to limited experimental resources, we conducted experiments on several machines with different computing environments. The primary settings and their roles on each machine are as follows:

- **Local 1**: A single machine with Ubuntu 20.04 LTS, one Intel i9-10920X CPU with 24 cores, two NVIDIA GeForce RTX 3090 GPUs with 24GB VRAM each, 251GiB RAM, Python 3.10, and CUDA version 12.0. On this machine, we conducted RQ1, extracted the fine-tuning dataset, executed TDE attacks on reference and target LMs, and analyzed experimental results.

- **Local 2**: A single machine with Ubuntu 20.04 LTS, one Intel i9-11900F CPU with 16 cores, one NVIDIA GeForce RTX 3090 GPU with 24GB VRAM, 125GiB RAM, approximately 25TB of storage (2TB SSD and 23TB HDD), Python 3.10, and CUDA version 12.2. Here, we reconstructed the pre-training dataset and computed true positives.

- **Cloud (Naver Smart Machine Learning; NSML)** (Sung et al., 2017): A single machine with Ubuntu 20.04 LTS, one Intel Xeon Gold 5220 CPU with 16 cores, two NVIDIA V100 GPUs with 32GB VRAM each, 176GiB RAM, Python 3.8, and CUDA version 11.4. We extracted the fine-tuning dataset on this machine and executed TDE attacks on reference and target LMs. On this machine, we performed PPO fine-tuning for OPT models other than OPT-6.7B and 13B.

- **Cloud (Kakao i Machine Learning; KiML)**[4]: A single machine with Ubuntu 18.04 LTS, one AMD EPYC 7763 CPU with 64 cores (only 16 cores available), one NVIDIA A100

---

[4] https://docs.kakaoi.ai/ml/

Table 3: Details of the reconstructed pre-training datasets of OPT. "Tokens" refers to words tokenized by the OPT tokenizer from each dataset. "Examples" denotes the number of data included in each training dataset. "Binary" represents the size after converting the dataset into a binary file. "Suffix Array" indicates the size after converting the binary file into a suffix array. "Total" is the sum of the sizes of the binary and suffix array files.

| Dataset | Tokens | Examples | Binary (GB) | Suffix Array (GB) | Total (GB) |
|---|---|---|---|---|---|
| BookCorpus | 1,084,248,174 | 74,006,376 | 2.8 | 11.0 | 13.8 |
| USPTO | 9,956,527,973 | 11,123,493 | 20.0 | 100.0 | 120.0 |
| Project Gutenberg | 3,066,013,103 | 28,601 | 6.1 | 30.7 | 36.8 |
| OpenSubtitles | 5,464,176,068 | 632,493 | 10.9 | 54.7 | 65.6 |
| Wikipedia | 12,054,096,912 | 16,940,015 | 24.2 | 121.2 | 145.4 |
| DM Mathematics | 7,388,934,036 | 1,918,563 | 14.8 | 74.0 | 88.8 |
| HackerNews | 2,069,555,419 | 1,571,990 | 4.2 | 16.6 | 20.8 |
| Pile-CC | 53,187,982,906 | 52,442,909 | 106.8 | 534.1 | 640.9 |
| OpenWebText2 | 16,414,911,293 | 17,103,564 | 33.0 | 131.9 | 164.9 |
| Reddit | 264,176,333,658 | 6,024,114,846 | 564.5 | 2712.5 | 3277.0 |
| **Total** | 374,862,779,542 | 6,199,882,850 | 787.3 | 3786.7 | 4574.0 |

GPU with 80GB VRAM, 128GiB RAM, Python 3.8, and CUDA version 11.4. On this machine, we performed PPO fine-tuning for OPT-6.7B.

- **Cloud (Vast.ai)**[5]: A single machine with four NVIDIA A100 GPUs with 80GB VRAM each, Python 3.10, and CUDA version 11.7. We conducted PPO fine-tuning for OPT-13B on this machine.

All experiments across these machines used the PyTorch (Paszke et al., 2019) (v2.0), HuggingFace Transformers (Wolf et al., 2020) (latest version), and DeepSpeed (Rasley et al., 2020) (v0.10) libraries. We also referred to pre-implemented code from DeepSpeedExamples (Yao et al., 2023) for fine-tuning.

## B.2 Pre-training Dataset Reconstruction

As mentioned in §5.1, to verify the results of the TDE attack, we reconstructed 10 out of the 12 datasets used in the OPT pre-training. This process consists of two steps: (1) downloading the datasets and converting them into binary files, and (2) generating a suffix array (Manber & Myers, 1993). We referred to the GitHub repository[6] that implemented EXACTSUBSTR by Lee et al. (2022) to generate the suffix array. Table 3 shows the detailed specifications of the reconstructed pre-training datasets.

## B.3 Fine-tuning Dataset Creation

In §4.1, we described how to create a fine-tuning dataset with pseudo-labeled membership. Several texts were dropped among the 100,000 generated texts for various reasons: (1) "Span" that the generated text was composed of fewer than the predetermined threshold of 20 words, (2) "NaN" that occurred during the calculation of the log-likelihood of perturb texts for perturbation discrepancy, and (3) "Odd" that due to the previous two reasons, there remained an odd number of generated texts, causing a remainder when matching in pairs. We specify the number of texts dropped for these reasons in Table 4. Given the relatively small proportion of dropped samples to the total number of generated texts, we believe that dropping these samples will have minimal impact on fine-tuning. Generally, as the model size increases, the number of dropped texts tends to decrease, and we speculate this is because larger models generate high-quality texts.

Subsequently, in Table 5, we report the fine-tuning dataset size used in §4.2, excluding the dropped generated texts. First, we match the generated texts in pairs and divide them into 80% training and 20% evaluation datasets. Each dataset is then split again into 40% for RM and 60% for the target LM fine-tuning.

---

[5] https://vast.ai/
[6] https://github.com/google-research/deduplicate-text-datasets

Table 4: The number of samples dropped from the initial 100,000 texts generated by each model.

| OPT | Span | NaN | Odd | Total |
|---|---|---|---|---|
| 125M | 29 | 0 | 1 | 30 |
| 350M | 50 | 10 | 0 | 60 |
| 1.3B | 1 | 18 | 1 | 20 |
| 2.7B | 2 | 13 | 1 | 16 |
| 6.7B | 2 | 0 | 0 | 2 |
| 13B | 2 | 0 | 0 | 2 |
| **Total** | 86 | 41 | 3 | 130 |

Table 5: The number of elements in all datasets used for fine-tuning RM and target LM.

| OPT | RM Data | | PPO Data | | Total | |
|---|---|---|---|---|---|---|
| | Split | Size | Split | Size | Split | Size |
| 125M | train | 15,995 | train | 23,993 | train | 39,988 |
| | eval | 3,999 | eval | 5,998 | eval | 9,997 |
| | total | 19,994 | total | 29,991 | total | 49,985 |
| 350M | train | 15,990 | train | 23,986 | train | 39,976 |
| | eval | 3,998 | eval | 5,996 | eval | 9,994 |
| | total | 19,988 | total | 29,982 | total | 49,970 |
| 1.3B | train | 15,997 | train | 23,995 | train | 39,992 |
| | eval | 3,999 | eval | 5,999 | eval | 9,998 |
| | total | 19,996 | total | 29,994 | total | 49,990 |
| 2.7B | train | 15,997 | train | 23,996 | train | 39,993 |
| | eval | 4,000 | eval | 5,999 | eval | 9,999 |
| | total | 19,997 | total | 29,995 | total | 49,992 |
| 6.7B | train | 16,000 | train | 23,999 | train | 39,999 |
| | eval | 4,000 | eval | 6,000 | eval | 10,000 |
| | total | 20,000 | total | 29,999 | total | 49,999 |
| 13B | train | 16,000 | train | 23,999 | train | 39,999 |
| | eval | 4,000 | eval | 6,000 | eval | 10,000 |
| | total | 20,000 | total | 29,999 | total | 49,999 |

## B.4 FINE-TUNING REWARD MODEL

This section describes the hyperparameters for training the RM with the pseudo-labeled dataset prepared in Appendix B.3. We referred to the examples in DeepSpeedExamples (Yao et al., 2023) for fine-tuning. All RMs used the pre-trained OPT-350M. For RM training, we employ Zero Redundancy Optimizer 3 (ZeRO-3) Offload (Ren et al., 2021). The training mini-batch size is set to 32; the batch size is a multiplication of per device batch size, the number of GPUs, and gradient accumulation size. Therefore, we have the total optimization steps of 500 iterations. The learning rate is $5 \times 10^{-5}$, and we use a cosine annealing scheduler (Loshchilov & Hutter, 2017) without a warm-up step. Weight decay is set at $0.1$.

In Figure 5, we report the training log of the RM for each OPT model based on the derived dataset. The log is publicly available online[7].

---

[7]https://tensorboard.dev/experiment/cdi2xqDGR3mey3opQu7Idw/

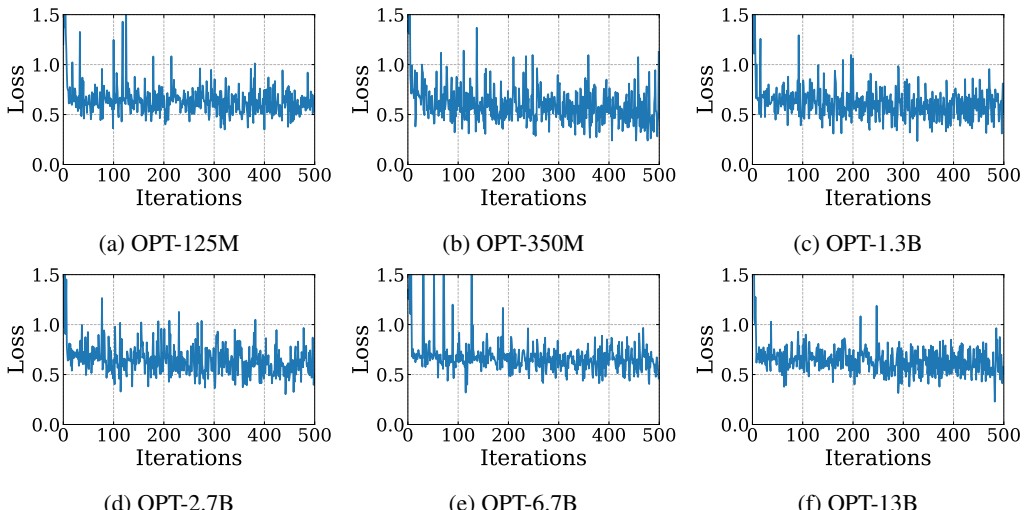

Figure 5: Training logs of RM for the dataset generated by each OPT model.

## B.5 FINE-TUNING TARGET LANGUAGE MODEL

This chapter describes the hyperparameters for training the target LM using the pseudo-labeled dataset prepared in Appendix B.3. Similarly to the RM, we referred to the examples in DeepSpeedExamples for fine-tuning. We used ZeRO-3 Offload for both the training of the actor—*i.e.*, the target model— and the critic—*i.e.*, the reward model. The training mini-batch size for all models except OPT-13B is fixed at 32; the batch size is a multiplication of per device batch size, the number of GPUs, and gradient accumulation size. Since renting GPU resources is expensive, we increased the training batch size for OPT-13B to 64 for fast optimization. Therefore, the total optimization step is 750 iterations. The learning rate for actor training is consistently $9.65 \times 10^{-6}$ regardless of model size, and the learning rate for critic training is $5 \times 10^{-6}$, both using a cosine annealing scheduler with 100 warm-up steps. Weight decay was not used. We commonly applied a 128-dimensional LoRA (Hu et al., 2022) to the actor model.

In Figure 6, we report the training log of the target LM according to the dataset derived from each OPT model. Due to constraints in the experimental environment, we omitted the training log for OPT-6.7B. The log can be checked publicly online[8].

## C FURTHER EXAMPLES

### C.1 PERTURBED TEXTS

Table 6 shows an example of perturbing a single generated sentence using the mask-and-fill approach. For specific methods, refer to §4.1.

### C.2 PSEUDO-LABELED TEXTS

Table 7 shows examples of two matching generated sentences labeled as chosen and rejected. For specific methods, refer to §4.1.

### C.3 MEMORIZED CONTENTS

Table 8, Table 9, Table 10, and Table 11 show some samples memorized by OPT-1.3B (USPTO), OPT-1.3B (Pile-CC), OPT-2.7B (OpenWebText), and OPT-2.7B (Reddit), respectively. We have masked some sensitive information, such as email addresses.

---

[8] https://tensorboard.dev/experiment/j6PkVA5lS96X0n1pygo2sQ/

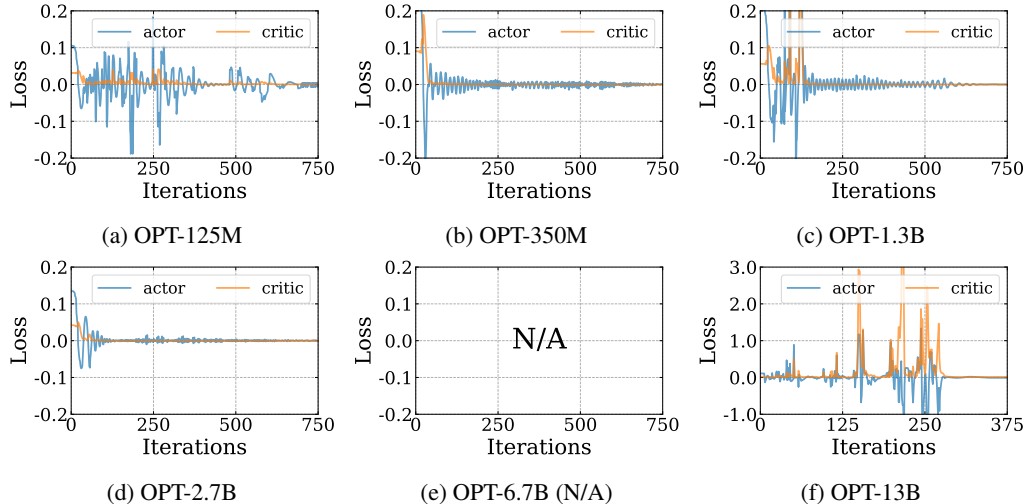

Figure 6: Training logs of the target LM trained with datasets generated by each OPT model. We omitted the training log for OPT-6.7B.

Table 6: Examples of creating perturbed text for each generation. After selecting two consecutive spans from the generated text (red), we replace them with a single mask (blue) and predict the corrupted span (green).

| **Generated Text** |
| --- |
| I can't watch the video right now, but this looks very interesting to me. Is it as fun as it looks?\nFrom the reviews, it's a good port. The PC version is apparently more stable, but the game is still very enjoyable. Edit: and it has native 60 fps.\nThank you for that info. I love this game, but I haven't been able to get it to work on windows.\nIt's still in EA but the developer is very active. If you really want to play the game with all the fixes the developer has to offer, then it's your best bet.\nCool. I'm gonna give it a try. I've always just played it on my Mac with bootcamp. Thanks!\nI just tried it out. The sound stuttered but not as much as on the emulator. Not as playable for me. I also couldn't get all the settings to work that are supposed to help performance on the PC version. :(\nThat sucks. I know there are performance issues, and I'm not surprised at all. But I've been really enjoying it on the go.\nYeah, I enjoyed it. I just have to wait for the PC port to be fixed I guess. But the |

| **Masked Text** |
| --- |
| I can't watch the video right now, but this looks very interesting to me. Is it as fun as it [MASK1] reviews, it's a good port. The PC version is apparently more stable, but the [MASK2] still very enjoyable. Edit: and [MASK3] native 60 fps.\nThank you for [MASK4] I love this game, but I haven't been able to get it to work on windows.\nIt's still in EA but the developer is very active. If you really want to play the game with all the fixes the developer has to offer, then it's your best [MASK5] gonna give it a try. I've always just played it on my Mac with bootcamp. Thanks!\nI just tried [MASK6] The sound stuttered but not as much as on the emulator. Not as playable [MASK7] I also couldn't get all the settings [MASK8] that are supposed to help performance on the PC version. :(\nThat sucks. I know there are performance issues, and I'm not surprised at [MASK9] I've been really enjoying it on the go.\nYeah, I enjoyed it. I just have to wait for the PC port to be fixed [MASK10] But the |

| **Perturbed Text** |
| --- |
| I can't watch the video right now, but this looks very interesting to me. Is it as fun as it looks? According to reviews, it's a good port. The PC version is apparently more stable, but the Mac version is still very enjoyable. Edit: and it's running at native 60 fps.\nThank you for this. I love this game, but I haven't been able to get it to work on windows.\nIt's still in EA but the developer is very active. If you really want to play the game with all the fixes the developer has to offer, then it's your best bet. I'm gonna give it a try. I've always just played it on my Mac with bootcamp. Thanks!\nI just tried it. The sound stuttered but not as much as on the emulator. Not as playable though. I also couldn't get all the settings adjusted that are supposed to help performance on the PC version. :(\nThat sucks. I know there are performance issues, and I'm not surprised at all. I've been really enjoying it on the go.\nYeah, I enjoyed it. I just have to wait for the PC port to be fixed . But the |

Table 7: Pseudo-labeled generations examples. We matched texts with calculated perturbation discrepancies in pairs, labeling the text with the lower value as chosen and the one with the higher value as rejected.

| **Chosen Text** (perturbation discrepancy: −1.81) |
|---|
| It is a long list, but I will only highlight one of the most salient points about the list. It is a point that many people, in all walks of life, seem to be unaware of. I have noticed over the years that many leaders of businesses and other organizations tend to assume that they are the best leaders, and that everyone else, especially the employees, should just follow them. I think many leaders forget that every company has a culture and a history.\n\nYou can't have one without the other.\nMany of the leaders of the major companies in America are graduates of Harvard Business School.\nAnd Harvard is known for being extremely selective.\nThere are, of course, other schools that also teach business, but Harvard is by far the most expensive,\nAnd since they don'tsave all of their profits from tuition, they have to charge high prices.\nThe average price of a full course of studies at Harvard is a whopping $70,000!\n\nI don't know if you are aware of this, but there are two other schools in the United States that are equally selective in selecting students. One is Stanford, the other is MIT. These schools charge far less than Harvard. Their average price is about |

| **Rejected Text** (perturbation discrepancy: 0.86) |
|---|
| How did you make the bread part?\nWith yeast\nI meant to say gmore specifically :P\nI didn't! I'm guessing you use sourdough starter. I made a starter a while back using a recipe that came off the internet. I forget the exact measurements but I think it's flour, salt, water and some starter.\nWhat kind of flour?\nAny flour! We bought all purpose and it turned out fine!\nDid you knead? It usually takes me an hour and a half for a batch like that.\nNo I did not knead. It'd be nice to try that one day though. That sounds like a good use for an entire day in the kitchen.\nIt's a very fun thing to do. I just had to make like 7 of these for a festival this weekend and they took forever.\nSounds like it! I wish we had a festival nearby. Where do you live?\nI live in Chicago but I was in a suburb of Cleveland. We're in the middle of a major food festival right now and there's still a lot of baking to be done. I've got two batches of dough that need to be made in the next |

Table 8: An example from USPTO memorized by the OPT-1.3B.

| **Generated Text** |
|---|
| ⋯ In addition, when an external data is to be decoded, there is a need for increased time for decoding the data, and an error rate can be increased. Thus, there will be a problem that it is hard to increase data capacity at the same time.\nThe above information disclosed in this Background section is only for enhancement of understanding of the background of the invention and therefore it may contain information that does not form the prior art that is already known in this country to a person of ordinary skill in the art.\nIt is a primary object of the present invention to provide a dynamic random access memory (DRAM) capable of increasing data capacity by solving the problems described above. ⋯ |

| **Original Example** |
|---|
| ⋯ However, the related art does not consider the state of the auxiliary battery, so that the life of the auxiliary battery may be decreased, and further, the auxiliary battery may be discharged in a case in which the load of a vehicle is sharply changed.\nThe above information disclosed in this Background section is only for enhancement of understanding of the background of the invention and therefore it may contain information that does not form the prior art that is already known in this country to a person of ordinary skill in the art. ⋯ |

Table 9: An example from Pile-CC memorized by the OPT-1.3B.

| **Generated Text** |
| --- |
| · · · This article was automatically imported from our old content management system. If you see any display errors, please let us know: ██@cadence11.ca.\n\nView all current articles from our community\n\nPostmedia wants to improve your reading experience as well as share the best deals and promotions from our advertisers with you. The information below will be used to optimize the content and make ads across the network more relevant to you. You can always change the information you share with us by editing your profile.\nBy clicking "Create Account", I hearby grant permission to Postmedia to use my account information to create my account.\nI also accept and agree to be bound by Postmedia's · · · |

| **Original Example** |
| --- |
| · · · We are using Facebook commenting. Visit our FAQ page for more information.\n\nAlmost Done!\n\nPostmedia wants to improve your reading experience as well as share the best deals and promotions from our advertisers with you. The information below will be used to optimize the content and make ads across the network more relevant to you. You can always change the information you share with us by editing your profile.\n\nBy clicking "Create Account", I hearby grant permission to Postmedia to use my account information to create my account. · · · |

Table 10: An example of OpenWebText memorized by the OPT-2.7B.

| **Generated Text** |
| --- |
| \n\nThe former president has been silent since he left office in January. He spoke in July at the funeral of Dr. Martin Luther King Jr. and in April, on the second day of this year's Republican National Convention.\n\nMr. Obama had also previously spoken at the April 2004 funeral of civil rights leader Rev. Martin L. King Jr., who was gunned down in 1968.\n(ABC News)\n\nThis site contains copyrighted material the use of which has not always been specifically authorized by the copyright owner. We are making such material available in our efforts to advance understanding of environmental, political, human rights, economic, democracy, scientific, and social justice issues, etc. We believe this constitutes a 'fair use' of any such copyrighted material as provided for in section 107 of the US Copyright Law. In accordance with Title 17 U.S.C. Section 107, the material on this site is distributed without profit to those who have expressed a prior interest in receiving the included information for research and educational purposes. For more information go to: http://www.law.cornell.edu/uscode/17/107.shtml. If you wish to use copyrighted material from this site for purposes of your own that go beyond 'fair |

| **Original Example** |
| --- |
| · · · \n\nLet your Friends know! Be sure to SHARE!\n\nFair Use Notice:This site contains copyrighted material the use of which has not always been specifically authorized by the copyright owner. We are making such material available in our efforts to advance understanding of environmental, political, human rights, economic, democracy, scientific, and social justice issues, etc. We believe this constitutes a 'fair use' of any such copyrighted material as provided for in section 107 of the US Copyright Law. In accordance with Title 17 U.S.C. Section 107, the material on this site is distributed without profit to those who have expressed a prior interest in receiving the included information for research and educational purposes. For more information go to: http://www.law.cornell.edu/uscode/17/107.shtml. If you wish to use copyrighted material from this site for purposes of your own that go beyond 'fair use', you must obtain permission from the copyright owner. · · · |

Table 11: An example of Reddit memorized by the OPT-2.7B.

| **Generated Text** |
| --- |
| a lot of attention," said the source. "She just can't seem to get a break, which is sad. Her life is hard and she deserves better."\n\nWe pay for juicy info! Do you have a story for RadarOnline.com? Email us at ██@radaronline.com, or call us at (866) ON-RADAR (███-███) any time, day or night.\n\nGet the exclusive insider's story from Hollywood's hottest stars by subscribing to our new podcast Straight Shuter below! · · · |

| **Original Example** |
| --- |
| · · · "\n\nDo you think Griffith is on a downward spiral? Tell us in the comments! We pay for juicy info! Do you have a story for RadarOnline.com? Email us at ██@radaronline.com, or call us at (866) ON-RADAR (███-███) any time, day or night." · · · |

