# OpenReview forum: "Amplifying Training Data Exposure through Fine-Tuning with Pseudo-Labeled Memberships"
_ICLR.cc/2024/Conference — Submitted to ICLR 2024_

### Official Review · Reviewer_cSHP · 2023-10-30

**Soundness:** 2 fair
**Presentation:** 3 good
**Contribution:** 3 good
**Rating:** 6
**Confidence:** 3

**Summary:**

The paper presents a novel attack strategy aimed at increasing the vulnerability of pre-trained language models to training data extraction attacks. By adversarially fine-tuning the LMs, the authors claim to amplify the exposure of sensitive pre-training data. They propose the use of pseudo-labels to help fine-tune the model in a way that favors text likely to have originated from the pre-training dataset. Their experiments suggest that this approach can lead to a significant increase in training data exposure, particularly in large models with over 1 billion parameters.

**Strengths:**

1. Generally, the paper is well-written and easy to follow.
2. The paper introduces a unique and unexplored attack vector that goes beyond the traditional post-hoc data extraction methods.
3. Given the widespread use of large LLMs, the paper addresses a timely and significant issue of data privacy.

**Weaknesses:**

1. The paper assumes the availability of restricted white-box capabilities, which may not always be the case in real-world scenarios.
2. Although the author provided extensive empirical study results, I'm still curious about the underlying mechanism behind the attack. Could the author elucidate how the proposed adversarial fine-tuning method effectively amplifies data exposure? It might be helpful to use a naive linear classification task as an illustrative example.

**Questions:**

How would the efficacy of this adversarial fine-tuning approach change if some form of differentially private training was already applied to the pre-trained language model? Would the attack still be as effective, or would it require significant modifications?

---

> ### Author Response · Authors · 2023-11-14
> **Response to Reviewer cSHP**
>
> Thank you for acknowledging our main contributions:
>
> - well-written and easy to follow.
> - introducing a unique and unexplored attack vector.
> - addressing a timely and significant issue of data privacy.
>
> ---
>
> > [Q1] How would the efficacy of this adversarial fine-tuning approach change if some form of differentially private training was already applied to the pre-trained language model? Would the attack still be as effective, or would it require significant modifications?
>
> [Q1] Unlike previous approaches that aim to leak more effective training data from LMs, our method primarily targets making the LM itself more vulnerable. From this perspective, we anticipate that our approach would still be effective—*i.e.*, amplifying the risk of leakage several times—even in LMs with well-defined defense strategies like differentially private training applied (Section 6).
>
> ---
>
> > [W1] The paper assumes the availability of restricted white-box capabilities, which may not always be the case in real-world scenarios.
>
> [W1] The restricted white-box scenario assumed in our study is often considered unrealistic in previous research [1] and thus has not been explored yet. However, this assumption is increasingly essential for the following reasons (details in Section 3.1):
>
> 1. **Increase in Public LMs**: Efforts towards open science are leading to an increase in publicly available LMs, which pose potential risks related to our assumption.
> 2. **Risk of LM Weight Leakage**: Sophisticated black-box model extraction attacks make our assumption more realistic.
> 3. **Customized LM API Services**: Recently, services that support customizing back-end LMs by fine-tuning them with user datasets are emerging (e.g., OpenAI [2]), and our research could be a severe concern in such cases.
>
> Specifically, item 3 is not mentioned in our paper, so we plan to enhance the content by adding this in the final version.
>
> ---
>
> > [W2] Although the author provided extensive empirical study results, I'm still curious about the underlying mechanism behind the attack. Could the author elucidate how the proposed adversarial fine-tuning method effectively amplifies data exposure? It might be helpful to use a naive linear classification task as an illustrative example.
>
> [W2] Due to difficulties in explaining our approach using a linear classifier as an example, we will provide an explanation along with an example of an adversarial attack on ResNet to describe the primary mechanism of our attack.
>
> The primary mechanism of our approach that effectively amplifies exposure involves increasing the probability of the LM generating responses that are expected to contain more (partial) training data by rewarding such responses more highly. Our approach is akin to performing 'gradient ascent' on a ResNet model to make it misclassify more adversarial examples, thereby becoming increasingly vulnerable to adversarial attacks.
>
> | Target Model | Risk | Evaluation Metric | Result | Social Impact |
> | --- | --- | --- | --- | --- |
> | ResNet (example) | Misclassification | Accuracy | Vulnerable ResNet to Adversarial Attacks | Low |
> | OPT (ours) | Training Data Exposure | Machine-generated Probability (i.e., perturbation discrepancy) | Vulnerable OPT to Training Data Extraction Attacks  | High (due to privacy risks) |
>
> ---
>
> Thank you for the constructive comments. We hope these explanations resolve the concerns. Further questions or suggestions would also be appreciated.
>
> [1] Carlini, Nicholas, et al. "Extracting training data from large language models." *30th USENIX Security Symposium (USENIX Security 21)*. 2021.
>
> [2] https://platform.openai.com/docs/guides/fine-tuning

---

> > ### Comment · Reviewer_cSHP · 2023-11-22
> >
> > Thank you, author(s), for your response. Regarding Q1, I believe it would be beneficial to include some experiments to substantiate your argument. As for Q2, I appreciate your willingness to enhance the details in support of your white-box settings. Concerning Q3, I understand your point about the ResNet example, and while it may not be the most suitable illustration, it's acceptable if a linear classifier explanation is unavailable. I will maintain my original score.

---

> ### Author Response · Authors · 2023-11-22
> **Response to Reviewer cSHP**
>
> Thank you for your response. Regarding the answer to Q1, we believe that experiments in robust LMs applied with possible mitigations (*e.g.*, differentially private (DP) training) can demonstrate the effectiveness of our approach, thereby enhancing its contributions. While we positively consider these aspects in our subsequent work, we would like to point out the difficulty of this DP training from scratch due to limited experimental resources. If you could advise us about any public-available defense models we might be missing (possibly named something like DP-GPT), it would be greatly helpful in setting our future direction. Regarding the answers to Q2 and Q3, we once again express our appreciation for your constructive comments.

---

### Official Review · Reviewer_JBPh · 2023-10-31

**Soundness:** 2 fair
**Presentation:** 3 good
**Contribution:** 2 fair
**Rating:** 5
**Confidence:** 4

**Summary:**

This paper focuses on amplifying training data memorization in terms of the extraction attack performance. The goal is to put the target model in a state where it is more likely to regurgitate training data.

More specifically, the authors propose a fine-tuning method, using reinforcement learning, text generation and machine-text generation detection to condition the model such that it is likelier to regurgitate its training data. They do this with restricted whit-box access to the model, and no access to the training data. They attempt to achieve this by doing the following steps: 1) generating many samples from the model 2) using detectgpt to give scores on how likely each generation is to be human written, intuition being that human written text is more likely to have been pre-training data. 3) create pairs of training/generation data 4) training a reward model to distinguish between the generation and pseudo training data. 5) fine-tune the target model using the reward model. 6) taking samples from the new model and comparing tot the non-trained reference model.

The authors then test the performance of the proposed method by taking samples from the fine-tuned model and then measuring exact matches with training data and reporting the values. They compare these numbers to those of a non-fine tuned model. They also study the performance of the reward model separately.

**Strengths:**

1. The approach/way of looking at the extraction problem is novel, prior work usually focuses on coming up with post-hoc extraction and not fine-tuning-based methods, where the decoding process is modified such that it incentivizes training data extraction. This paper however, tries to change the model so that its more likely to generate training data.

2. The problem is also an important problem, as current extraction methods are not very successful, most of them demonstrating low extraction rates.

**Weaknesses:**

1. Lack of enough experiments and ablations to support the main claim of the paper, that the method amplifies memorization. See questions 1-3 below. This is my main concern with the approach, as the model might as well just be regurgitating the same set of n-grams, over and over and as the reported is not measured over deduplicate generations based on n-grams (it seems like the only deduplication performed is wrt to full matched strings with training data), nor is there a diversity metric reported. Intuitively, I would assume that the fine-tuning is going to get the model to collapse on the set of generations used for RM training/FT. I also wonder why the authors did not use a metric similar to BLEU.

2. The structure of the paper is hard to follow and its not really well written. Some of the results are not explained well, also the way the deduplication is performed is not fully clear.  See questions 3 and 4 below.

3. Section 5.2 only shows how well a reward model can learn to differentiate between machine generated and human written text. It does not provide any evidence to support the claims of the paper regarding training data. It is simply an ablation. I am not sure what it is included as one of the first results. The fact that the reward model can differentiate between different texts does not necessarily translate to it being better at incentivizing the target model to regurgitate training data.

**Questions:**

1. Can the authors disentangle how much of the extractions overlap with the generated text that they fine-tuned with, and how much of the extracted text is non-overlapping and actually a generation of the model that is due to the amplification. Right now the main remaining question is does this method actually reinforce memorizations or is it just overfitting to the pseudo labeled data? (this corresponds to weakness 1 from above)

2. How many of the generated samples after fine-tuning differ from the reference model generations? as mentioned in question 1, if the model is collapsing, the new generations that the fine-tuned model has would overlap a lot with the generations from the refenrece model. it would be interesting to see if that is the case, or if there are any entirely new generations.

3. One main problem with the results is that the generation deduplication is happening on a full string level, and not n-gram overlaps. Same as point 1, I think there is probably huge overlap, what is the diversity of generations? There are no ablations here. We need a lot more ablations on the experiments.

4.  section 5.2 please elaborate on the duplicate token overlaps, and the intervals. I went over the text multiple times but did not realize what the point of that experiment is.

---

> ### Author Response · Authors · 2023-11-14
> **Response to Reviewer JBPh (1/2)**
>
> Thank you for acknowledging our main contributions:
>
> - a novel approach to extraction attack.
> - a possibility of potential improvements to current extraction methods with low extraction rates.
>
> ---
>
> > [Q1] Can the authors disentangle how much of the extractions overlap with the generated text that they fine-tuned with, and how much of the extracted text is non-overlapping and actually a generation of the model that is due to the amplification. Right now the main remaining question is does this method actually reinforce memorizations or is it just overfitting to the pseudo labeled data? (this corresponds to weakness 1 from above)
>
> [Q1] In the following table, we report the true positives after deduplicating some texts containing actual training data from reference and fine-tuned LMs. Here, 'duplication' refers to cases where 'two different generated texts are extracted from the same training data point.' As the adversary in Section 3.1's assumption does not consider the duplication of extracted texts (Section 3.1), we did not perform separate n-gram-based deduplication [3].
>
> | OPT | ● | ○ | Inc. |
> | --- | --- | --- | --- |
> | 125M | 56 | 139 | $\times$2.5↑ |
> | 350M | 128 | 272 | $\times$2.1↑ |
> | 1.3B | 74 | 621 | $\times$8.4↑ |
> | 2.7B | 96 | 425 | $\times$4.4↑ |
> | 6.7B | 130 | 477 | $\times$3.7↑ |
> | 13B | 165 | 425 | $\times$2.6↑ |
>
> We observed that, even after deduplication, there is an amplification of 2.6-8.4 times in LMs with over 1B parameters. While some performance decreases in the 13B parameter model, our adversarial fine-tuning enhances TDE attack performance up to eight times. We will add these ablation studies in the final version.
>
> ---
>
> > [Q2] How many of the generated samples after fine-tuning differ from the reference model generations? as mentioned in question 1, if the model is collapsing, the new generations that the fine-tuned model has would overlap a lot with the generations from the refenrece model. it would be interesting to see if that is the case, or if there are any entirely new generations.
>
> [Q2] We report in the following table how much overlap exists between samples extracted from the fine-tuned LM and those from the reference LM. Each extracted sample has not been deduplicated (i.e., the total number is the same as in Table 2), and each range represents the number of overlapping tokens.
>
> | OPT | [0,64) | [64,128) | [128,192) | [192,256) | 256 | Total | % (<64) |
> | --- | --- | --- | --- | --- | --- | --- | --- |
> | 125M | 100 | 46 | 18 | 5 | 0 | 169 | 59.17 |
> | 350M | 158 | 70 | 38 | 59 | 0 | 325 | 48.62 |
> | 1.3B | 545 | 83 | 30 | 117 | 0 | 775 | 70.32 |
> | 2.7B | 505 | 12 | 0 | 0 | 0 | 517 | 97.68 |
> | 6.7B | 534 | 11 | 0 | 0 | 0 | 545 | 97.98 |
> | 13B | 704 | 81 | 1 | 0 | 0 | 786 | 89.57 |
>
> These experimental results show that generations extracted from our fine-tuned LM usually do not overlap with those not fine-tuned. We set the 'overlap' threshold to 25% (*i.e.*, 64 tokens), and70.32% to 97.98% of extraction results from fine-tuned LMs with over 1B parameters are unique. Based on your feedback, we will add these ablation studies to the final version.
>
> ---
>
> > [Q3] One main problem with the results is that the generation deduplication is happening on a full string level, and not n-gram overlaps. Same as point 1, I think there is probably huge overlap, what is the diversity of generations? There are no ablations here. We need a lot more ablations on the experiments.
>
> [Q3] To check the diversity of total generated texts (note, not extracted results) from both reference and fine-tuned LMs, we measured self-BLEU scores [1] and unique n-gram percentages [2]. First, the following table describes the stochastic self-BLEU scores.
>
> | OPT | ● | ○ | Dec. (%) |
> | --- | --- | --- | --- |
> | 125M | 0.33 | 0.26 | 20.74 |
> | 350M | 0.29 | 0.27 | 6.02 |
> | 1.3B | 0.28 | 0.24 | 15.80 |
> | 2.7B | 0.27 | 0.24 | 11.95 |
> | 6.7B | 0.27 | 0.23 | 15.22 |
> | 13B | 0.26 | 0.24 | 9.23 |
>
> Our fine-tuning strategy decreased self-BLEU scores by 6.02% to 20.74%, showing improved diversity across all parameter ranges. Next are the results for unique n-gram percentages for n=2, 3, 4.
>
> |  | n=2 |  |  | n=3 |  |  | n=4 |  |  |
> | --- | --- | --- | --- | --- | --- | --- | --- | --- | --- |
> | OPT | ● | ○ | Diff. | ● | ○ | Diff. | ● | ○ | Diff. |
> | 125M | 12.91 | 18.69 | 1.45 | 40.31 | 49.74 | 1.23 | 70.84 | 78.03 | 1.10 |
> | 350M | 15.43 | 19.64 | 1.27 | 44.91 | 50.22 | 1.12 | 74.62 | 77.56 | 1.04 |
> | 1.3B | 16.14 | 26.95 | 1.67 | 46.64 | 58.36 | 1.25 | 76.13 | 80.74 | 1.06 |
> | 2.7B | 17.31 | 23.45 | 1.35 | 48.47 | 55.59 | 1.15 | 77.40 | 81.15 | 1.05 |
> | 6.7B | 17.66 | 22.37 | 1.27 | 49.06 | 54.75 | 1.12 | 77.93 | 81.60 | 1.05 |
> | 13B | 18.40 | 22.70 | 1.23 | 49.92 | 55.31 | 1.11 | 78.45 | 81.06 | 1.03 |

---

> > ### Comment · Reviewer_JBPh · 2023-11-22
> >
> > I thank the authors for their response. I have one follow up question, 64 is a bit too high of number of tokens, this paper (http://arxiv.org/abs/2111.09509) shows that generations become novel/unique with n>10 for n-grams. I would be curious to see this table for n=10, or at least 20. I am still not convinced of the diversity of the text and that it is not regenerating the fine-tuning data, because the self-bleu number increases are also not fully convincing.

---

> ### Author Response · Authors · 2023-11-14
> **Response to Reviewer JBPh (2/2)**
>
> Similarly, we confirmed the improvement in unique n-gram percentages in all settings. Like the previous two ablation studies, we will add the results of this ablation study on generations' diversity to the final version, further enhancing our paper.
>
> ---
>
> > [Q4] Section 5.2 please elaborate on the duplicate token overlaps, and the intervals. I went over the text multiple times but did not realize what the point of that experiment is.
>
> [Q4] We will answer based on the overlap mentioned in Section 5.1. If our understanding is inaccurate, please let us know the specific position of a page, section, paragraph, or line so we can provide a more detailed response.
>
> Fundamentally, the metric for the effectiveness of our TDE attack is the true positives among 100,000 generated texts (Section 5.1). To determine whether a generation is part of the training data, we need to calculate how much it overlaps with a training data sample (in the training dataset) of the target LM. Since each generation has precisely 256 tokens, the unit of overlap is tokens (*i.e.*, not word or byte). Inspired by previous research, we set the number of overlapping tokens at 50 [3]. In summary, if 50 consecutive tokens appearing in a generation of the target LM exist at any location in the training dataset, we consider the generation to have been 'extracted' from the training dataset.
>
> We further segment the 'duplicate intervals' of generated texts with [50, 256] token overlaps to dive deeper into our result. We divide the true positive samples into five duplicate intervals as follows (Table 2): [50, 64), [64, 128), [128, 192), [192, 256), {256}.
>
> ---
>
> > [W1] Lack of enough experiments and ablations to support the main claim of the paper, that the method amplifies memorization. See questions 1-3 below. This is my main concern with the approach, as the model might as well just be regurgitating the same set of n-grams, over and over and as the reported is not measured over deduplicate generations based on n-grams (it seems like the only deduplication performed is wrt to full matched strings with training data), nor is there a diversity metric reported. Intuitively, I would assume that the fine-tuning is going to get the model to collapse on the set of generations used for RM training/FT. I also wonder why the authors did not use a metric similar to BLEU.
>
> [W1] Through your questions 1 to 3, we conducted additional ablation studies to demonstrate that our strategy amplifies memorization. Specifically, we confirmed that (1) our fine-tuning does not compromise the diversity of LM's generated texts (Q3), and (2) even with deduplication of texts extracted from the same training data position, the amplification remains still effective (Q1). These results suggest that our fine-tuning is less likely to lead to LM collapse.
>
> ---
>
> > [W2] The structure of the paper is hard to follow and its not really well written. Some of the results are not explained well, also the way the deduplication is performed is not fully clear. See questions 3 and 4 below.
>
> [W2] We will add extensive ablation studies to the experiment and clarify the meaning of 'duplicate token overlaps,' improving the paper's structure.
>
> ---
>
> > [W3] Section 5.2 only shows how well a reward model can learn to differentiate between machine generated and human written text. It does not provide any evidence to support the claims of the paper regarding training data. It is simply an ablation. I am not sure what it is included as one of the first results. The fact that the reward model can differentiate between different texts does not necessarily translate to it being better at incentivizing the target model to regurgitate training data.
>
> [W3] We agree with your opinion that the experiment in Section 5.2 (Table 1) does not support the paper's primary claim and is somewhat close to an ablation study. We acknowledge that the experiment observing whether the reward model can distinguish samples that seem 'more/less machine-generated' and whether such distinguishability varies with the scale of the target LM has a relatively trivial contribution. We plan to improve the structure of the forthcoming paper for an enhanced presentation.
>
> ---
>
> Thank you for the constructive comments. We hope these explanations resolve the concerns. Further questions or suggestions would also be appreciated.
>
> [1] Zhu, Yaoming, et al. "Texygen: A benchmarking platform for text generation models." *The 41st international ACM SIGIR conference on research & development in information retrieval*. 2018.
>
> [2] Wang, Alex, and Kyunghyun Cho. "BERT has a mouth, and it must speak: BERT as a Markov random field language model." *arXiv preprint arXiv:1902.04094* (2019).
>
> [3] Carlini, Nicholas, et al. "Extracting training data from large language models." *30th USENIX Security Symposium (USENIX Security 21)*. 2021.

---

> ### Author Response · Authors · 2023-11-22
> **Response to Reviewer JBPh**
>
> Thank you for your follow-up question. We have measured the proportion of novel/unique n-grams in the generations from the fine-tuned LM, referencing the paper (in Figure 1) you raise (*i.e.*, the count of n-grams in the generations that did not appear in the fine-tuning dataset in Table 5 of our manuscript).
>
> | OPT | Total 10-grams | Unique 10-grams | Proportion (↑) |
> | --- | --- | --- | --- |
> | 125M | 24,334,096 | 24,306,061 | 0.9988 |
> | 350M | 24,129,396 | 24,096,257 | 0.9986 |
> | 1.3B | 23,572,674 | 23,548,392 | 0.9990 |
> | 2.7B | 24,175,634 | 24,148,747 | 0.9989 |
> | 6.7B | 24,428,210 | 24,404,788 | 0.9990 |
> | 13B | 24,156,194 | 24,125,084 | 0.9987 |
>
> Similarly to the paper (in Figure 1) you raise, the generations from the fine-tuned LM contained over 99% of the 10-gram set being novel/unique. We believe these results serve as an indicator of the diversity in the generations of the fine-tuned LM. We will cite the prior work and include this ablation study in the final version of our manuscript. We hope these explanations resolve the concerns.

---

### Official Review · Reviewer_ZqEj · 2023-11-01

**Soundness:** 3 good
**Presentation:** 3 good
**Contribution:** 3 good
**Rating:** 6
**Confidence:** 3

**Summary:**

The paper proposes a new attack strategy to increase the exposure of private training data from pre-trained language models. The main contributions are:
1. The paper introduces a novel scenario where an attacker fine-tunes a pre-trained language model with self-generated texts that are pseudo-labeled based on their machine-generated probabilities. The paper assumes that texts with lower machine-generated probabilities are more likely to contain training data.
2. The paper uses a zero-shot machine-generated text detection method (DetectGPT) to calculate the perturbation discrepancy of each generated text, and a reinforcement learning from human feedback method (RLHF) to fine-tune the target language model to favor texts with lower perturbation discrepancy.
3. The paper evaluates the proposed attack strategy on six versions of the OPT language model and shows that it can amplify the training data exposure by four to eight times compared to the reference models. The paper also analyzes the extracted samples and discusses potential mitigations and future research directions.

**Strengths:**

1. Originality: The paper introduces a novel attack scenario where an adversary fine-tunes a pre-trained language model to amplify the exposure of its training data. This strategy differs from prior studies by aiming to intensify the model’s retention of its pre-training dataset. The paper also proposes a two-step approach to achieve this goal, involving pseudo-labeling based on machine-generated probabilities and reinforcement learning with self-generations. To the best of my knowledge, this is the first work to explore such an attack strategy and demonstrate its feasibility and effectiveness.
2. Quality: The paper is well-written and provides sufficient technical details and empirical evidence to support its claims. The paper follows the standard structure of an ICLR submission and adheres to the formatting guidelines. The paper also discusses potential mitigations and countermeasures against the proposed attack, as well as open questions for future research. The paper uses appropriate references and citations to acknowledge previous work and situate its contribution in the literature.
3. Clarity: The paper is clear and easy to follow. The paper defines the threat model, the adversary’s capabilities and objective, and the main steps of the attack strategy in a precise and coherent manner. The paper also explains the rationale and intuition behind each step of the attack, as well as the challenges and assumptions involved. The paper uses figures, tables, and equations to illustrate the key concepts and results. The paper also provides qualitative analysis of extracted samples and discusses the limitations and implications of the attack.
4. Significance: The paper addresses an important and timely problem of training data extraction attacks on neural language models, which pose serious privacy risks for both data owners and model users. The paper demonstrates that such attacks can be amplified by adversarial fine-tuning, which can increase the exposure of sensitive training data by up to eight times. The paper also provides insights into the factors that affect the vulnerability of language models to such attacks, such as model size, training dataset type, and perturbation function. The paper contributes to advancing the understanding and mitigation of privacy threats in language modeling.

**Weaknesses:**

1. The paper does not specify how the adversary evaluates the effectiveness of the TDE attack, and what are the assumptions and limitations of the attack scenario. The paper also does not compare or contrast its attack strategy with existing TDE attacks in terms of feasibility, scalability, and practicality.
2. The paper relies on a single zero-shot machine-generated text detection method (DetectGPT) to pseudo-label the self-generated texts, without considering other possible methods or evaluating the robustness and reliability of DetectGPT. The paper also does not explain how the perturbation discrepancy correlates with the membership probability or the presence of training data in the generated texts. The paper does not account for the potential confounding factors or sources of bias in its experiments, such as the choice of prompts, sampling methods, hyperparameters, datasets, and evaluation metrics.
3. The paper does not discuss the ethical and social implications of its attack strategy. The paper proposes a novel form of TDE attack that can amplify the exposure of sensitive and private information from pre-trained LMs, but does not address the potential harms or risks that such an attack can pose to individuals, organizations, or society at large.

**Questions:**

1. In Figure 1, perturbed LM generations are divided into two classes: "good answer" and "bad answer," based on the value of d(x). Was the threshold for d(x) chosen empirically?
2. In Table 1 for Epoch 1, the three values with the lowest test accuracy are highlighted. In contrast, for Epoch 2, the highlighted values represent the top-3 highest test accuracy. There are no highlights in Epoch 0 and Epoch 3. Should the highlighting approach be consistent, or was this variation done intentionally for a specific reason?

---

> ### Author Response · Authors · 2023-11-14
> **Response to Reviewer ZqEj (1/3)**
>
> Thank you for acknowledging our main contributions:
>
> - a novel attack scenario where an adversary fine-tunes a pre-trained language model to amplify the exposure of its training data.
> - well-written and provides sufficient technical details and empirical evidence to support its claims.
> - clear and easy to follow.
> - advancing the understanding and mitigation of privacy threats in language modeling.
>
> ---
>
> > [Q1] In Figure 1, perturbed LM generations are divided into two classes: "good answer" and "bad answer," based on the value of d(x). Was the threshold for d(x) chosen empirically?
>
> [Q1] Instead of setting a separate threshold to distinguish between good and bad answers for d(x), we decide based on their relative magnitude. We paired 100,000 generations into 50,000 pairs and, within each pair, pseudo-labeled the generation with the lower and higher d(x) as the good and bad answer, respectively. Consequently, we end up with 50,000 good answers and 50,000 bad answers. Please refer to Section 4.1 for more details.
>
> ---
>
> > [Q2] In Table 1 for Epoch 1, the three values with the lowest test accuracy are highlighted. In contrast, for Epoch 2, the highlighted values represent the top-3 highest test accuracy. There are no highlights in Epoch 0 and Epoch 3. Should the highlighting approach be consistent, or was this variation done intentionally for a specific reason?
>
> [Q2] In Table 1, we highlighted the epoch with the highest classification accuracy 'for each model' (i.e., the highest value in each row is displayed). In the final version, we will enhance readability by adding shading to the rows in the table to aid the reader's understanding.
>
> ---
>
> > [W1-a] The paper does not specify how the adversary evaluates the effectiveness of the TDE attack, and [W1-b] what are the assumptions and limitations of the attack scenario. [W1-c] The paper also does not compare or contrast its attack strategy with existing TDE attacks in terms of feasibility, scalability, and practicality.
>
> [W1-a] The evaluation metric for the effectiveness of an adversary's TDE attack is the true positive (Section 5.2). Specifically, the adversary generates 100,000 texts containing precisely 256 tokens from the target LM and then reports the number of generations extracted from the training dataset. In this case, we consider a generation to be 'memorized (or extracted)' if exactly 50 consecutive tokens in the generation are present in the training data.
>
> [W1-b] As you pointed out, our research assumes a restricted white-box scenario. The restricted white-box scenario assumed in our study is often considered unrealistic in previous research [1] and thus has not been explored yet. However, this assumption is increasingly essential for the following reasons (details in Section 3.1):
>
> 1. **Increase in Public LMs**: Efforts towards open science are leading to an increase in publicly available LMs, which pose potential risks related to our assumption.
> 2. **Risk of LM Weight Leakage**: Sophisticated black-box model extraction attacks make our assumption more realistic.
> 3. **Customized LM API Services**: Recently, services that support customizing back-end LMs by fine-tuning them with user datasets are emerging (e.g., OpenAI [3]), and our research could be a severe concern in such cases.
>
> Specifically, item 3 is not mentioned in our paper, so we plan to enhance the content by adding this in the final version.

---

> ### Author Response · Authors · 2023-11-14
> **Response to Reviewer ZqEj (2/3)**
>
> [W1-c] As feasibility and practicality are already discussed in [W1-a] and Section 3.1, we will address the remaining concern—scalability. In this paper, instead of demonstrating experimental results across various architectures (*e.g.*, GPT-Neo-X, LLaMA, and BLOOM), we focused on experiments with different model parameters within a single architecture, OPT. This decision was based on the following reasons:
>
> 1. **Consistency in Recent LM Architectures**: We noted that most recent LMs are commonly based on the Transformer decoder architecture. Therefore, we judged that showing the applicability of our approach across various architectures would not significantly enhance the contribution of our research.
> 2. **Limited Experimental Resources**: Due to constraints in resources such as GPUs, storage, and workforce (Section B.1), we conducted our experiments by reducing the diversity of model architectures within a range that did not compromise the validity of our approach.
>
> We believed that due to the first reason, reporting the performance of our work using just the OPT family was sufficient. While the limitations mentioned in the second reason restricted the scope of this paper, we will note the possibility of applications to other architectures like BERT or T5 in our paper as promising future work.
>
> ---
>
> > [W2-a] The paper relies on a single zero-shot machine-generated text detection method (DetectGPT) to pseudo-label the self-generated texts, without considering other possible methods or evaluating the robustness and reliability of DetectGPT. [W2-b] The paper also does not explain how the perturbation discrepancy correlates with the membership probability or the presence of training data in the generated texts. [W2-c] The paper does not account for the potential confounding factors or sources of bias in its experiments, such as the choice of prompts, sampling methods, hyperparameters, datasets, and evaluation metrics.
>
> [W2-a] The primary reason for adopting DetectGPT in this study is its state-of-the-art (SOTA) performance and its capability to calculate machine-generated probability in a zero-shot manner. To maximize the robustness and reliability of DetectGPT, we intentionally matched generations with significant perturbation discrepancies during the pairing process (Section 4.1). We believe this local optimal pseudo-labeling process significantly mitigates the issues you pointed out.
>
> [W2-b] In this study, we assume that the lower the perturbation discrepancy, the lower the machine-generated probability [2], and therefore, the higher the likelihood of the generations being human-written (Section 4.1). We will reexamine the coherence of this statement in the text and enhance it in the final version.
>
> [W2-c] We did not conduct a specific search for the optimal combination of hyperparameters, such as the sampling method, during our study (Section 5.1). The main reason is that the adversary can independently combine our adversarial fine-tuning approach with previous TDE attacks. We agree with your concern about not using all training data sets of the target LM for verification or potential bias due to the choice of evaluation metrics could be an issue. In the final version, we will add a 'Limitations' Section to address these concerns comprehensively.
>
> ---
>
> > [W3] The paper does not discuss the ethical and social implications of its attack strategy. The paper proposes a novel form of TDE attack that can amplify the exposure of sensitive and private information from pre-trained LMs, but does not address the potential harms or risks that such an attack can pose to individuals, organizations, or society at large.
>
> [W3] As you mentioned, we are concerned about the potential harm and risk factors associated with the public disclosure of our research results. To address these issues, we plan to add an 'Ethics and Broader Impacts' Section in the final version. Briefly, the potential risks identified are as follows:
>
> 1. **Unfair Competitive Practices**: If private training data is used in the training of an LM, leaking such information could be considered an act of unauthorized appropriation of data collected, refined, and stored with a significant investment of time and resources by a company. Furthermore, an adversary could attempt to deduce the training data preprocessing methods from the attributes and properties of the refined training dataset.
> 2. **Personal Information Leakage**: The vast training data used in recent LMs may include unfiltered or sensitive data in context [4]. If an adversary extracts such sensitive information from the target LM, it could lead to serious privacy breaches.
>
> ---
>
> Thank you for the constructive comments. We hope these explanations resolve the concerns. Further questions or suggestions would also be appreciated.

---

> ### Author Response · Authors · 2023-11-14
> **Response to Reviewer ZqEj (3/3)**
>
> [1] Carlini, Nicholas, et al. "Extracting training data from large language models." *30th USENIX Security Symposium (USENIX Security 21)*. 2021.
>
> [2] Mitchell, Eric, et al. "Detectgpt: Zero-shot machine-generated text detection using probability curvature." *arXiv preprint arXiv:2301.11305* (2023).
>
> [3] https://platform.openai.com/docs/guides/fine-tuning
>
> [4] Brown, Hannah, et al. "What does it mean for a language model to preserve privacy?." *Proceedings of the 2022 ACM Conference on Fairness, Accountability, and Transparency*. 2022.

---

### Official Review · Reviewer_VkQN · 2023-11-01

**Soundness:** 3 good
**Presentation:** 2 fair
**Contribution:** 3 good
**Rating:** 6
**Confidence:** 2

**Summary:**

This work investigates how model fine-tuning may potentially make the models more vulnerable to leaking their pre-train dataset. The authors apply the machine-generated text with more like human-written to fine-tune the language models. Reinforcement learning with self-generation is employed to fine-tune the models. To demonstrate the effectiveness of their approach, the author conducts experiments on six datasets over 6 language models with different amounts of trainable parameters.

**Strengths:**

1. This work proposes a new perspective to make data extraction attacks on pre-training language models easier.
2. This study performs experiments across diverse datasets and various models, enhancing the generalizability of the empirical analysis.

**Weaknesses:**

1. While the author explores various models in the experiments, there is a noticeable lack of diversity in their architectures; all the studied models originate from the same architectural family.
2. It would be valuable if the authors could show some qualitative results, e.g., reconstructed text in the model fine-tuning with their approach and the standard approaches.
3. There is no model utility performance comparison between this work and the other work.

**Questions:**

see weakness.

---

> ### Author Response · Authors · 2023-11-14
> **Response to Reviewer VkQN**
>
> Thank you for acknowledging our main contributions:
>
> - a new perspective for training data extraction attacks.
> - experimenting across various datasets and models to enhance the generalizability.
>
> ---
>
> > [W1] While the author explores various models in the experiments, there is a noticeable lack of diversity in their architectures; all the studied models originate from the same architectural family.
>
> [W1] In this paper, instead of demonstrating experimental results across various architectures (*e.g.*, GPT-Neo-X, LLaMA, and BLOOM), we focused on experiments with different model parameters within a single architecture, OPT. This decision was based on the following reasons:
>
> 1. **Consistency in Recent LM Architectures**: We believe that showing the performance of our approach in different architectures would not significantly enhance the contribution of this study, as recent LMs generally use the Transformer decoder-based architecture. Following your valuable suggestions, we will mention the application to other structures like BERT and T5 as promising future work in our paper. We will gladly consider your advice for expansion in our future research.
> 2. **Limitation of Experimental Resources**: Due to the limitations in experimental resources like GPU, storage space, and workforce (as mentioned in Section B.1), we conducted our experiments with the OPT family only, reducing the diversity of model architectures without compromising the validity of our approach.
>
> ---
>
> > [W2] It would be valuable if the authors could show some qualitative results, e.g., reconstructed text in the model fine-tuning with their approach and the standard approaches.
>
> [W2] We understood your mention of 'standard approaches' as 'standard TDE attack' (*i.e.*, without fine-tuning the model). As we can add the qualitative results you mentioned in time for the final submission, we will enhance the final version by comparing the differences in reconstructed texts between LMs with and without fine-tuning.
>
> ---
>
> > [W3] There is no model utility performance comparison between this work and the other work.
>
> [W3] As emphasized in Sections 3.1 and 6, the adversary possesses a replica of the target LM, thus eliminating the need to consider a decline in the model's utility, like validation perplexity. Also, as this is the first study amplifying LM exposure through adversarial fine-tuning, we used the TDE attack results on the reference LM as a baseline (in Table 2) instead of comparing performance with other works.
>
> ---
>
> Thank you for the constructive comments. We hope these explanations resolve the concerns. Further questions or suggestions would also be appreciated.

---

### Author Response · Authors · 2023-11-21
**A kind reminder regarding our response**

Dear Reviewer,

As the ICLR rebuttal period is approaching its end, we kindly remind you to review our submitted response. Your feedback is essential for finalizing our work.

Thank you for your attention.

Best regards,

The Authors

---

> ### Author Response · Authors · 2023-11-23
> **Remarks and Appeciation**
>
> Dear Reviewers,
>
> As the rebuttal period draws to a close, we sincerely hope our response, crafted based on your constructive comments, aids in furthering a positive evaluation of our work. We are grateful for the opportunity to refine our submission with your insights.
>
> Thank you for your thoughtful review and time.
>
> Best regards,
>
> The Authors

---

### Meta-Review · Area_Chair_UhuM · 2023-12-05

**Metareview:**

The authors propose a new type of setting, where the goal is to amplify training data exposure - e.g. make trainin data easier to extract with subsequent membership inference attacks. They do this by fine-tuning the data on pretraining dataset-like data, using LLM-generated-text detetors.

Strength: most reiewers agree that this is a novel approach to an important problem.

Weakness: the experimental validation seems questionable (JBPh), relies on LLM text detection methods that are often known to be unreliable (ZqEj)

**Justification For Why Not Higher Score:**

core issues like experimental design and reliance on zero-shot detectors without careful testing and validation are problems that probably should be resolved.

**Justification For Why Not Lower Score:**

N/A

---

### Decision · Program_Chairs · 2024-01-16

Reject